# Priority Health Needs and Challenges in New Zealand Pacific Communities—A Qualitative Analysis of Healthcare Delivery during the COVID-19 Pandemic

**DOI:** 10.3390/healthcare11162239

**Published:** 2023-08-09

**Authors:** Ravi Reddy, John Sluyter, Atefeh Kiadarbandsari, Malakai Ofanoa, Maryann Heather, Fuafiva Fa’alau, Vili Nosa

**Affiliations:** 1College of Health, Massey University, Private Bag 11 222, Palmerston North 4442, New Zealand; r.reddy@massey.ac.nz; 2Faculty of Medical and Health Sciences, University of Auckland, Auckland 1023, New Zealand; j.sluyter@auckland.ac.nz (J.S.); akia647@aucklanduni.ac.nz (A.K.); m.ofanoa@auckland.ac.nz (M.O.); m.heather@auckland.ac.nz (M.H.); f.faalau@auckland.ac.nz (F.F.)

**Keywords:** public health, vulnerable communities, equity, pandemic, cultural responsiveness

## Abstract

Background: The Pacific community in New Zealand experienced an increased risk of COVID-19 transmission due to delayed contact tracing, along with a disproportionate prevalence of health challenges. The community is representative of a diverse population who proudly identify with the vibrant Pacific Island nations of Samoa, Tonga, Cook Islands, Niue, Fiji, etc. Pacific communities in New Zealand face a higher burden of health challenges compared to other groups. These challenges include obesity, high blood pressure, diabetes, mental health disorders, respiratory issues, smoking, excessive alcohol consumption, disabilities, and chronic conditions. Concerns were raised regarding the oversight of Pacific community views in the initial pandemic response planning. Pacific healthcare professionals expressed concerns about inadequate state support and the need for active involvement in decision making. Methods: This article reports thematic analyses of text data gained from open-ended questions from a purposive anonymous online survey completed by Pacific healthcare professionals in New Zealand. Results: The participants shared their experiences and opinions, which generated four major themes highlighting priority health needs and challenges. These themes included the necessity for a culturally appropriate healthcare plan, adequate resourcing, addressing discrimination, and emphasising a united and collaborative effort for consistency. The research’s limitation is the narrow scope of open-ended questions in the questionnaire survey. However, conducting semi-structured face-to-face interviews can provide more in-depth data and offer further insights beyond the four broad themes identified in the analysis. Conclusions: The findings can inform the development of future research to provide more in-depth data and offer further insights beyond the four broad themes identified in the analysis. This will help develop future tailored healthcare delivery plans that address specific Pacific community needs.

## 1. Introduction

The COVID-19 response significantly affected various aspects of people’s lives due to the precautionary measures implemented by governments. These measures included testing and contact tracing, quarantines, social distancing requirements, remote work arrangements, and national lockdowns. Prioritising the wellbeing of the nation against the deadly virus was accepted in the early days of the pandemic without much resistance in many countries [1]. The New Zealand government implemented an elimination strategy against COVID-19, incorporating control measures like contact tracing, border closure, national lockdowns, extended school and work closures, and activity restrictions [2]. Governments that effectively communicated pandemic response strategies could navigate public perception and gain strong public support, but risked public backlash and potential replacement if they failed to take timely and decisive actions [3]. New Zealand was one of the first countries to adopt a stringent elimination strategy based on a four-tier Alert Level system that was reinforced by the “Go Hard, Go Early” message [4]. The effectiveness of a country’s approach to COVID-19 appeared to also depend on effectively managing and balancing the social and political impacts, as well as meeting the needs and expectations of individuals and communities [5].

It has been reported that minority communities experienced greater disparities and inequitable access to healthcare services during the pandemic [6,7]. For example, the speed at which telemedicine or video consultations were rolled out meant that communities and systems were not sufficiently knowledgeable or adequately resourced to use it effectively [8]. Telehealth holds promise, especially for those confident in the use of technology, but presents additional challenges for groups with specific language and cultural needs [9]. Therefore, it is essential to develop tailored awareness campaigns and policies to address the unique needs of vulnerable groups, such as Pacific communities in New Zealand. The Pacific community forms 8.1% of the New Zealand population and consists of a vibrant and culturally diverse group of individuals who have connections to Polynesia, Micronesia, and Melanesia in the South Pacific [10]. There is evidence that Pacific communities faced an elevated risk of COVID-19 transmission due to delayed contact tracing, in comparison to other ethnicities [11]. 

Even prior to the pandemic, Pacific communities in New Zealand experienced a disproportionate prevalence of various health challenges. These challenges included obesity, high blood pressure, diabetes, mental health disorders, respiratory issues, smoking, excessive alcohol consumption, disabilities, and chronic conditions [12]. Pacific people are more likely to live in high-deprivation areas with low median household incomes, higher unemployment rates, the lowest rates of home ownership, and the highest rates of household overcrowding [13]. During the COVID-19 pandemic, a Pacific patient, aged 54.7 years, without any reported comorbidities, had a comparable risk of hospitalisation to an 80-year-old patient in the New Zealand European/Other group without any reported comorbidities [14]. Pacific communities also faced higher infection fatality rates resulting from COVID-19 [15] and Pacific healthcare professionals voiced concerns regarding various aspects of the pandemic response with respect to Pacific communities [16]. They have reported inadequate early support from health organisations and the need for clearer and more supportive guidance in terms of communication and decision making [17,18]. Despite improvements in clinical efforts, such as improved vaccination rates within Pacific communities, there was initially a lack of adequate social support for essential needs like food, shelter, and resources [16]. This long-standing concern has been advocated by Pacific health leaders even before the pandemic. However, the challenges were overcome by recognising the pivotal role of Pacific cultural values in targeted public campaigns and awareness and adherence to the COVID-19 elimination strategy within the Pacific community [10]. Under immense pressure, healthcare systems relied on dedicated professionals working tirelessly on the frontline. Pacific healthcare service providers united, mobilising resources and extended their reach to support Pacific communities [19]. 

The pandemic highlighted the limitations of the healthcare system and public health strategies, which existed even before COVID-19 [16]. It is important to identify what strategies were successful and what aspects may have fallen short and understand how these factors influenced health outcomes within the communities. Considering the limited presence of Pacific individuals within the healthcare workforce, accounting for approximately 2% [20], it was important to examine their viewpoints and experiences regarding the COVID-19 response in relation to the Pacific communities they serve. Therefore, the aim of this study was to examine the viewpoints of healthcare professionals of Pacific descent regarding the response to the pandemic. This will facilitate informing further comprehensive research based on the findings of this exploratory investigation. 

## 2. Materials and Methods

This was an exploratory study investigating the perspectives and experiences of Pacific healthcare professionals during the COVID-19 pandemic in New Zealand. A purposive anonymous online survey which included three open-ended questionnaires was facilitated between December 2020 and May 2021. The text-form data were qualitatively analysed and are reported in this article. 

### 2.1. Participants and Procedure

The inclusion criteria were that participants needed to be of Pacific descent and working in the health sector during levels 3 and 4 of the lockdowns. The research was open to potential participants of all genders, age groups, and places of location in New Zealand. The exclusion criteria were health professionals who did not identify with a Pacific Island ethnicity and Pacific health professional not working during levels 3 and 4 of the lockdowns. Purposive sampling was used to recruit the participants. The survey was advertised through email contacts via established Pacific community networks. Ethical information was delivered at the beginning of the questionnaire and respondents were requested to indicate their consent to participate before continuing with the survey. The English language survey completed through a Qualtrics (Qualtrics, Provo, UT, USA) link took approximately 10 min to complete. A total of 17 Pacific healthcare professionals participated in the study between December 2020 and May 2021. It should be noted that Pacific people are underrepresented in the health workforce, making up approximately 2% of the overall healthcare workforce [20]. Ethical approval to conduct the research was granted by the University of Auckland Human Participants Ethics Committee (reference #: 024697).

### 2.2. Data Collection

Participants could provide open-ended suggestions about what could have been done in response to the pandemic. Three questions were specifically developed to assess the Pacific healthcare professionals’ experiences during levels 3 and 4 of the lockdowns. These levels refer to a four-layer national guideline on restrictions established by the New Zealand government as a part of COVID-19 response strategies [2]. The questions were the following:
Is there anything new that could or should have been done differently?Do you suggest any changes to structures in healthcare access and primary health care for the response?How best could Pacific communities work together with others on a co-designed strategy to respond to a future pandemic?

Transcription was not required as textual data were collected. In addition, follow-up exploration of data collected was not undertaken.

### 2.3. Data Analysis

Thematic analysis [21] was used to examine responses to the open-ended questions. This included familiarisation with data content and the generation of codes that described features of the data. First, each data item within each question was read and re-read, and thought-provoking ideas were noted. Second, initial codes were generated through an inductive method (i.e., recognising meanings by a few words or expressions). Third, themes were constructed to shape possible groups of meanings. Fourth, potential themes were reviewed by three team members for accuracy (AK, RR, VN) and to make sure they aligned with the research question. Themes were refined and, in the fifth stage, the themes were named and described by thematic labels which led to a thematic map. The research team reviewed and edited themes and subthemes for suitability and labelling. We explored if other terms could offer better description of the themes. These themes and codes were validated through consensus of the research group.

## 3. Results

Seventeen Pacific healthcare professionals completed the questionnaire. The mean age was 44.2 (range: 23–61) years and there were twelve females and five males. The ethnic composition of the sample was Samoan (n = 10), Tongan (n = 2), Niuean (n = 2), Cook Islander (n = 2), and Fijian (n = 1). Work roles were ten medical doctors, three nurses, one psychologist, one social worker, and two other healthcare roles. Eleven participants were from Auckland (six from South Auckland), two were from Waikato, and one each from Wellington, Christchurch, Otago, and Taupo. All healthcare professionals worked during level 3 or 4 restrictions.

### 3.1. Findings

Four themes were generated, demonstrative of the healthcare professionals’ experiences and opinions regarding the priority health needs and issues that challenged them during the pandemic. The four major themes were (1) a need to develop a culturally appropriate healthcare plan; (2) adequate resourcing; (3) addressing discrimination; and (4) a united and collaborative effort for consistency.

#### 3.1.1. Need to Develop a Culturally Appropriate Healthcare Plan

Culturally relevant and responsive planning and strategies are essential to address the specific challenges and barriers faced by Pacific communities.

“*Culturally relevant and responsive planning, strategies that reflect/acknowledge the existing issues/barriers faced by Pacific communities and can keep an eye for reducing inequalities whilst also making provisions for the safety, care and positive health of each community*”.

“*I think more specialised services for around Auckland would have been great. For South Auckland, Pacific people need more support with their spiritual and familial needs. I think the government didn’t take into consideration how lonely our people would be with family or church connections. Especially if it’s an older generation who are illiterate in technology literacy*”.

“*Co-design idea helpful. Working within your local community is a helpful start*”.

#### 3.1.2. Embrace Learning from This Experience

A subtheme of learning from the pandemic experiences and utilising them as guidelines to create a more comprehensive response plan emerged. This could be used to achieve clear and coordinated decision making among stakeholders.

“*Take lessons from the pandemic and use these as implementation guidelines to develop a more comprehensive response plan*”.

“*Recognise mistakes/lessons learnt from COVID-19 pandemic, consultation across all relevant sectors, e.g., Health, MSD [Ministry of Social Development], MBIE [Ministry of Business, Innovation and Employment] should have Pacific specific plans, engage with NGOs [Non-government organisations], church groups, youth who know communities well*”.

#### 3.1.3. Adequate Resourcing

There was a perception that ensuring sufficient resources during the COVID-19 response is crucial. This includes sufficient provision of personal protective equipment and vaccinations, improving virtual consultation and technology needs, staffing capability and capacity, more support for Pacific communities, and allocating adequate funding to effectively combat the pandemic.

“*Better supply of PPE and vaccinations to the frontline staff*”. “*More mobile services and provisions for remote patient care and monitoring should be something that we consider to resource better*”. 

“*Extra registered/trained staff for facilities with COVID positive patients; instead of using students to fill gaps. Not just business as usual understaffing*”.

“*GP/Nurse Prescriber was onsite for Level 4. Then subsequent lockdowns we haven’t had one, they added value to the team and also gave greater access to community*”.

“*Primary care needs extra support including access to workforce & funding to implement virtual care services and outreach community services*”.

#### 3.1.4. Addressing Discrimination

There was a view that efforts should be made to address and rectify any instances of discrimination that may arise during the response to the pandemic, to ensure equity and fairness to Pacific communities. 

“*Address gaps in technology, lack of digital devices, Wi-Fi etc discriminated Pacific communities, youth, students, workforce*”. 

“*Anything that increases the number of brown people and people who are culturally competent. Ensure policies and systems are anti-racist*”.

#### 3.1.5. A United and Collaborative Effort for Consistency 

The participants perceived that different Pacific organisations and communities could have worked better as a collective machinery to ensure consistency of efforts and messaging. There appears to be a perception that groups working separately caused confusion. 

“*There were too many PI [Pacific Island] groups involved…Just too many mixed messages out there. MOH [Ministry of Health] Pacific Team should have taken the lead right from the outset*”.

“*Form one leadership group*”, “*By having a central online hub for health workers and public to share information*” and “*Pooling of resources*”. 

“*By having a central online hub for health workers and public to share information*”.

“*Linking PHOs [Primary Health Organisations] in more closely with community-based groups*”.

## 4. Discussion

There were four themes that reflected the Pacific healthcare professionals perceptions of the COVID-19 response for Pacific communities. The themes were (1) a need to develop a culturally appropriate healthcare plan; (2) adequate resourcing; (3) addressing discrimination; and (4) a united and collaborative effort for consistency. To promote connectedness, solidarity, and trust within Pacific communities, it is recommended to involve Pacific leaders in the planning and execution of healthcare and future pandemic response plans [22]. Including Pacific leaders in these processes ensures that their perspectives and insights are considered. This will lead to more effective strategies that resonate with the community and foster a sense of trust and unity. This inclusion helps to build stronger relationships and better address the specific needs and concerns of Pacific communities during times of crisis. Imposing isolated bureaucratic decision-making processes on Pacific communities is culturally inappropriate and exacerbates the vulnerability of Pacific individuals during times of adversity, such as the ongoing pandemic [10]. The failure to consider the cultural context and specific needs of Pacific communities in these decision-making approaches results in an inability to address their unique challenges. In addition, it exacerbates the existing vulnerabilities faced by Pacific people. The importance of community engagement and cultural responsiveness has been strongly advocated to ensure healthcare service provision is equitable and not disadvantageous to Pacific communities. 

Engagement with Pacific communities could be approached through three levels of interaction. These levels include state-provided public resources, community mobilisation with social and cultural norms, and not-for-profit organisations acting as intermediaries between Pacific communities and the state [23]. This approach ensures effective communication of the specific perspectives and needs of Pacific communities. It allows for the allocation of resources as part of an effective healthcare and future pandemic response plan. However, this was missing from the early stages of the COVID-19 response as there were limited culturally and language-aware contact tracers available, which delayed engagement with Pacific communities [11]. A significant concern in high-income countries has been the passive participation of communities without decision-making powers during the pandemic [24]. This can be seen as tokenism and does not effectively address trust and confidence issues among disadvantaged groups in future response strategies.

The lack of meaningful involvement of Pacific communities in the development of COVID-19 response plans exposed the vulnerable to discrimination, leading to higher infection rates and lower vaccination rates within Pacific communities during the initial stages of the pandemic [25]. Nevertheless, the Pacific community witnessed a significant turnaround because of successful community mobilisation efforts in collaboration with healthcare providers [16]. This partnership played a pivotal role in enhancing vaccination rates and curbing infection rates among the Pacific population. A similar lack of meaningful inclusion and representation contributing to disparities in accessing essential information, healthcare services, and vaccination opportunities has been reported among Māori communities in New Zealand [26]. Pacific communities have endured racism and discrimination long before the onset of the pandemic [16]. The systemic discrimination has had a detrimental impact on the health outcomes of Pacific communities. It highlights the urgent need to address these disparities and ensure their active participation in the development and implementation of future response strategies. This finding does not only offer learnings for future crisis planning but also opportunities for improving the current healthcare service delivery for vulnerable groups like Pacific communities in New Zealand. 

This study highlights the importance of a unified and collective approach, emphasising that a single Pacific voice may provide greater benefits for Pacific communities during and after pandemics. A united Pacific front amplifies needs, concerns, and priorities while gaining recognition and responsiveness from policymakers, healthcare providers, and stakeholders. Private and public stakeholders, advocacy groups, effective communication methods, and partnership models play a crucial role in developing guidelines, digital health tools, healthcare restructuring, and patient support [8]. A unified voice can facilitate more effective advocacy, resource allocation, and policy development specifically tailored to address the unique challenges faced by Pacific communities. This finding underscores the significance of solidarity and collaboration within the Pacific community to achieve better healthcare service delivery outcomes. 

A limitation of this research is the narrow scope of the open-ended questions in the questionnaire survey. In addition, a larger sample size would have allowed for greater in-depth data analyses of additional themes. A lack of follow-up inquiry of the data analysed also restricted the richness of the findings. As discussed earlier, the Pacific health workforce represent approximately 2% of the total health workforce [20]. Given the low representation, a larger quantitative survey of items related to these broad themes and an in-depth interview design will offer further insights in future research. In addition, a more robust study design should include analyses of intra-community profiles such as gender, age, regional differences, socio-economic status, etc. Our study is limited in this respect, and we emphasise that the findings are not generalisable to be representative of the entire Pacific Island health workforce in New Zealand. The findings of this exploratory study establish the key overarching challenges faced by Pacific communities with respect to healthcare delivery, which could be used to design robust future research.

## 5. Conclusions

The insights shared by the participants have highlighted four key themes that underscore the priority health needs and challenges faced by Pacific communities in New Zealand. These themes describe the importance of implementing culturally appropriate healthcare plans, ensuring sufficient resources, tackling issues of discrimination, and fostering a unified and collaborative approach to achieve consistency in healthcare delivery. The findings of this study emphasise the importance of early engagement with the Pacific community in developing healthcare delivery strategies and preparing for future pandemics. This approach ensures that community’s perspectives, needs, and cultural considerations are fully considered. This would involve establishing inclusive spaces for dialogue, involving community leaders, organisations, and individuals with lived experiences. The small sample size and narrow scope of questions did not allow for rich data collection, but the themes identified here can inform future semi-structured interview research designs. The broad findings form the initial framework for future research based on a comprehensive investigation of healthcare service delivery and improving preparedness for future pandemics and crises.

## Data Availability

The data presented in this study are available in article quotations reported in the results.

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
