# Peer review of "Priority Health Needs and Challenges in New Zealand Pacific Communities—A Qualitative Analysis of Healthcare Delivery during the COVID-19 Pandemic"

_healthcare, 2023, doi:10.3390/healthcare11162239_

Round 1

Reviewer 1 Report

To authors The authors of the manuscript report of the work entitled “Pacific healthcare professionals’ perspectives on the COVID-19 2 response in New Zealand” The manuscript is interesting, concise, well written, and reports an important healthcare issue for some communities, it reflects community health concerns and worth sharing, but I have some comments. Abstract: Line 12. For your readers, you might want to clarify or just introduce Pacific community, Pacific Healthcare, as a unique community, as body and as a region… it might be read as healthcare in the whole pacific. Also be straightforward on what challenges are disproportionately prevalent in healthcare compared to other regions. You mention these in the boy, but do here too to substantiate and grab the first impression of readers. This is just to help reader understand study region/community, not everyone is aware of. Add a limitation in the abstract Revise keyword, they’re in the title, they shouldn’t be Introduction Paragraphs are small, some state the same topic and could be joined. A big portion can be discussed in the context of health policy in the country and tailored delivery strategies towards that specific community. But the problem here is you’re mirroring the “pandemic” as an example which softens its priority in the context of all types of difficulties you list social, economic, remote regions…etc. The reader loses focus on the main objective as we go through the manuscript. The message gets a bit obscured with policy making it a bit difficult to understand what the main message is until I come across a phrase on pandemic. I think it is important to know but rephrase some sentences to focus in the context of the “pandemic”. Then at the end paragraph conclude with the need for a comprehensive reform on inclusive healthcare strategy; perhaps giving live examples on deteriorating health infra structure you state. L 47 48,50 this paragraph has good info, but first sentences can be well addressed by recommending the advancement but lack of training and education, health policy can be tailored to different communities. In addition, please organize your thoughts on the quality of healthcare deliverables (treatments, practitioners, clinics…etc) and pandemic control strategies like contact tracing which is a critically important element of control that clearly reflects on health administrative power to initiate tracing loci in community, schools, work…etc. This essentially goes back to your first two paragraphs where you mention all types of biocontainment protocol strategies including “contact tracing”, I suggest you join the first two paragraphs, and start the second one with “However, these measures were (hard, unplanned, inadequate, or any thing you think” for pacifics, then focus on how contact tracing was not done and why state the reasons why wasn’t done, is it community structure that makes it difficult, is it related health policy, make it clear. L 56 what challenges, I know what you mean but, please clarify and differentiate between healthcare delivery challenges and community challenges, this distinction is important for the context you made. L 56 to 54 and L 65 to 76 these two paragraphs have mixed information. You could separate health effects on the community and healthcare delivery problems like shortages put them each together to make it easy on the reader. Perhaps community health issues can go back to paragraph starting with L35. And collect Healthcare delivery issues separately. There is a mix. The last paragraph must be explicit and succinct, say clearly what efforts, whose effort. I think the last paragraph is not strong enough as to the issues presented Material and Method If cross sectional, state the dates and methods of stats. Exclusion criteria, age. A major concern is that only 17 professionals? In six months?, is this professional bodies? It makes sense but just 17 respondents would make it significant? At least where these respondents in different or remote regions which would at least each represent a whole different region. L127 write out or describe short forms if first time Result and discussion I think you could use these comments to slightly modify your discussion lines to address the research questions you stated. You referred to the pandemic, but you also built a completely different line in health policy, sightly distinguishing these point would send a good message. See below: The main issue here is relevancy of your title and where you ended up making themes none of them directly speaks to the current outbreak. All the themes are healthcare policy issues tailored to a specific community. Since you present Covid pandemic, you can use all of your themes but again tailor them to the Pandemic, biocontainment issues, preparedness programs, measures for a virus elevated virulence in weakened community …etc. Find a way to corporate pandemic specifically in your themes or loosen up the title to include potential similar outbreaks or health crisis, then slightly modify your themes to accommodate these inclusive points. In summary, please explain healthcare needs (existing challenge) and pandemic biocontainment and capacity building. These are two interchangeable but independent health issues. L135, then please include an age criteria in methods L136 n = should be italic The important question: where is your analysis result? How do you know or how the small number of respondents create significance in creating themes? What stats you used? Line 270 Conclusion should be sounder and strongly address the issue. Please indicate what is the status of the pandemic now. If declining then, correct the phrase in this line 270 Add a limitation in conclusion. Limited responders is another one 

Author Response

We have attached our response. 

Reviewer 2 Report

Reviewer consulted with a specialist in qualitative research methods. 

1. There is no correspondence between the title, aim/purpose, and survey questions. The purpose is very broad (to study “experience”). Besides, the first two questions of the questionnaires are more closed than open questions because they provoke a yes/no answer.

Is there anything new that could or should have been done differently?

Do you suggest any changes to structures in healthcare access and primary health care for the response?

2. Why was a survey questionnaire used as a data collection method instead of a semi-structured interview, which is much more suitable for obtaining inductive information? The topic is not specific enough to conduct such explanatory research prior to the interviews. Why were not interviews done right away?

“A limitation of this research is a narrow scope of the open-ended questions in the questionnaire survey. While, the analyses highlighted four broad themes, a semi-structured face-to-face interview will offer further insights from more in-depth data”.

3. “The English language survey completed through a Qualtrics (Qualtrics, Provo, UT) link took approximately 10 minutes to complete.” Was 10 minutes really enough to obtain material that adequately reflects the phenomenon under study?

4. Why was convenience sampling chosen instead of some form of purposive sampling, which would be much better to ensure the scientific rigour of the study? The selection criteria should have been more specific.

5. It is now well known that there is no single type of thematic analysis. Please clearly specify and justify which type of thematic analysis was used and how this type corresponded to the purpose of the study?

See more recent information and debate on thematic analysis:

Braun, V., Clarke, V. (2021a). One size fits all? What counts as quality practice in (reflexive) thematic analysis? Qualitative Research in Psychology, 18(3), 328-352.

Braun, V., & Clarke, V. (2021b). Can I Use TA? Should I Use TA? Should I Not Use TA? Comparing Reflexive Thematic Analysis and Other Pattern-Based Qualitative Analytic Approaches. Counselling and Psychotherapy Research, 21, 37-47.

Braun, V., Clarke, V. (2022). Conceptual and design thinking for thematic analysis. Qualitative Psychology, 9(1), 3–26.

Braun, V., Clarke, V. (2023a). Is thematic analysis used well in health psychology? A critical review of published research, with recommendations for quality practice and reporting. Health Psychology Review, 1-24.

Braun, V., Clarke, V. (2023b). Toward good practice in thematic analysis: Avoiding common problems and be(com)ing a knowing researcher, International Journal of Transgender Health, 24(1), 1-6.

6. Themes are described very briefly; they do not provide sufficient insight into the codes or sub-themes that fill the theme. The themes do not seem to produce new and original findings on a topic as widely researched as Covid-19 and the experience of healthcare workers.

Author Response

We attached our comments.  

Reviewer 3 Report

Line 132 - Sentence ends with an extra period

Line 168 - Suggestion to define in square brackets MSD and MBIE;  note that NGOs definition may not be need (well known)

Line 182 - Note GP - may not need square bracket definition

Line 200 - Suggestion to define in square brackets PI and MOH (like on line 205)

Round 2

Reviewer 1 Report

Dear Authors

The authors have responded to many comments, thank you. However, many other comments either made worse or were not addressed well. Among these comments there are serious issues and concerns that critical. This needs to be clearly addressed for the journal as a media for a scientific publication.

The following are some of the issues still remain:

Not much is added to design, introduction, conclusion, or discussion

R2 review, comments to Editor:

Title is misleading for the sample size and whole community name is touted in

Sample is not representative, limited

Many revisions were not properly taken care of, or not done, or incorrectly addressed

Example is the English and different paragraphing contents with same message or using continuous phrases in different paragraphs

The new additions made it look like public news criticism rather than a scientific publication, this is critical and might not be suitable as a scientific publication

Conclusion was not quite well supported because of the issue of paucity in sampling data

The authors did not respond well to the suggestions that the representatives from different groups must represent a significant number. What was suggested was that any single representative must call for the “group” which should have a large number in the community eg. 100s, 1000s, millions…etc. This was not clear in the manuscript; and therefore, dependance or a few individuals would ideally shape a theme for healthcare policy making. This is not supported anywhere and is a serious limitation

Some more issues

use at least 1.5 line space, font size 10? Follow journal

L14 Identified by

L23, this is quite concerning, please consider “representing….sections, directorates, regions…etc to substantiate this low sample issue” you can’t consider survey with only 17 professionals!!, these professionals must represent significantly higher numbers …representing (a number or descriptive number such as thousands, millions, whole population …etc, this is critical) in this community for eg. Otherwise your title will mislead, you implied a whole community, then survey only 17?, best is to assign each one of these as coming from a sector in community or alike. These groups you mention in L64 65 should be represented by these 17, or alternatively and most effectively do not mention 17, mention the names of groups represented instead. You can understand the risk. One of these groups must include a privileged groups, if any, from those you assume other than pacific communities

Line 50, please remove names, by the government is fine

L78 who faced, pls make that clear because there is a line with different group

L80, if you start with a Furthermore, it implies continuation to previous sentences, please either change the opening or join paragraphs

L80 L 83 it is not clear who are these, pacific communities or others? If you’re opening a new paragraph, it must stand alone, define in the first sentence who are “they”

L84 vaccination improved?

L85 remove the comma after beginning, this paragraph is very short, should join up

L96, again you start with “However, implying a continuous story” move up these paragraphs please, which challenges this is a new paragraph should present and define something

L98 about 3 to 4 line sentence is very long, elimination of what?, this paragraph is not complete and vague

L105 L106 is this a paragraph? This is touted here not sure where it goes, pls clarify

L107 117, this paragraph is over emphasized and unfortunately a sense of emotion is clear rather than scientific content despite referencing. Stating common issues in all countries instead of focusing what makes your community unique or your problems stand out.

The language worsened in revision unfortunately, some serious issues brought up by non professional paragraphing contents

Author Response

We have added our file.

Reviewer 2 Report

Manuscript is improved and can be accepted for the publication.

Author Response

We have added our file.

Round 3

Author Response

25 July 2023

The Editor

Healthcare Journal

Thank you for the opportunity to revise the manuscript following the third review. We want to thank the reviewer for their feedback and suggestions.

Title: Identifying healthcare delivery challenges faced by Pacific communities in New Zealand during the COVID-19 pandemic.

This is a very specific title for a nonspecific qualitative approach that you did not do, you did a brief descriptive analysis leading to four themes: things like:

“Qualitative analysis of healthcare delivery in New Zealand Pacific communities during the COVID-19 pandemic reveals four major themes high-lighting priority health needs and challenges.”

We have taken on board this suggestion and revised the title to now read as:

“Priority health needs and challenges in New Zealand Pacific communities – a qualitative analysis of healthcare delivery during the COVID-19 pandemic”.

A clear specific statement of the findings of this study, is not presented in the conclusion; instead, a general statement is described. Opening of conclusion is what everyone will look, it has to be well written, I think no need for discontinue-continue format with framing by commas. Sentences like: While the COVID-19 pandemic is declining, its experience….might work

We have re-written the conclusion as per your suggestions. Please find highlighted in the manuscript.

The long multi-lined long sentence style still prevails – we have shortened sentences where we could, and these are highlighted in the manuscript.

Used email so, coding audio into text is not an issue, but does it best represent, justifying in the method with a sentence may be a good idea. Usually involves follow up questions, was there any?

We have added the following information in the “data collection” sub-section, “Transcription was not required as textual data was collected. In addition, follow-up exploration of data collected was not undertaken”.  We have also added this in the limitation discussion.

Please visit Line 389 about funding

Thank you, we have removed the phrase, “Please add”.

Please visit Line, what do you mean “deemed” obtained, do you have informed consent? Or no

We have revised the statement, “Written Informed Consent was not required. Participants involved in the study read the information sheet and proceeded to participate in the online anonymous survey”.

What we meant by “deemed” to have consented was that by proceeding to participate in the survey, participants have given informed consent.